# Blinded by Language: Multimodal LLMs Underuse Their Vision Backbone

## Abstract

Multimodal language models (MLLMs) have recently emerged as versatile models that unify visual perception with language understanding. However, their performance across core vision tasks remains poorly characterized relative to the traditional vision backbones—on which they are built. In this work, we provide a systematic comparison of MLLMs and their underlying vision backbones across a diverse set of benchmarks. Our analysis reveals a consistent gap: *MLLMs underperform their own vision-backbone on perception tasks such as object recognition with deficits of 10-15% in accuracy.* On the other hand, MLLMs demonstrate considerable gains in reasoning-heavy tasks, such as counting and relational understanding, where language grounding provides complementary benefits. One reason for this discrepancy lies in the limitations of current evaluation practices. Unlike Vision-Language Models (VLMs), MLLMs are evaluated through open-ended text generation, making results more sensitive to formatting errors and instruction-following failures rather than core visual competence. Finally, to encourage research into the vision-capabilities of MLLMs we provide a reduced set of evaluations requiring modest resources while maintaining diagnostic value.

## 1 Introduction

MLLMs align a pretrained large language model (LLM) - with its robust text understanding and generation capabilities - with a vision backbone, enabling unified reasoning over both modalities. This paradigm has rapidly advanced the field, producing models that can not only answer visual questions with free-form explanations but also engage in multimodal dialogue Liu et al. [2023b, 2024a], Li et al. [2023b]. While the integration of LLMs has undoubtedly expanded the scope of multimodal models, it remains unclear what progress LLMs have brought to vision-centric understanding. This raises a fundamental question: *do MLLMs retain the same level of visual competence as their vision backbones, or does alignment with an LLM come at the expense of core-perception?*

Many benchmarks study vision capabilities via classification [Russakovsky et al., 2015, Recht et al., 2019, Idrissi et al., 2022, Hendrycks et al., 2021], attributes and relations of objects [Al-Tahan et al., 2024a, Dumpala et al., 2024, Thrush et al., 2022]. Recent efforts to benchmark multimodal model have relied on larger composite suites of benchmarks that span several tasks such as recognition, OCR, counting, visual question answering, and object attributes etc. [Yu et al., 2024, Liu et al., 2024b, Li et al., 2023a, Yue et al., 2023]. These new efforts however are not compatible with the prior generation of benchmarks designed for vision-language models, making comparisons to modern MLLMs challenging.

In this work, we unify evaluation protocols of MLLMs on vision-centric capabilities. By applying standardized, controlled evaluations, we provide a systematic comparison of MLLMs with their underlying vision backbones. Our evaluation spans 48 benchmarks (e.g., ImageNet-1k, CIFAR-100,

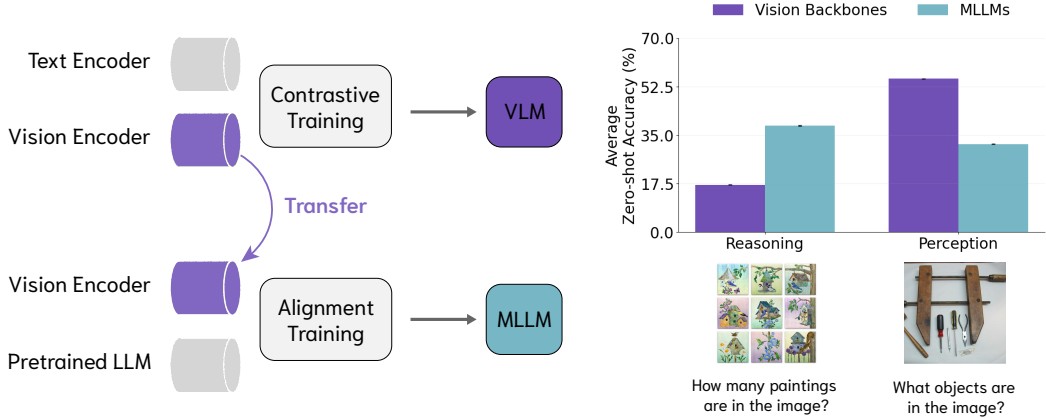

Figure 1: **VLMs and MLLMs exhibit complementary strengths: VLMs excel at perception, while MLLMs are stronger at reasoning.** VLMs are trained through contrastive learning between vision and text encoders, whereas MLLMs align pre-trained LLMs with the vision backbone inherited from VLMs. Benchmark comparisons illustrate that, despite reusing the same vision backbone, MLLMs shift performance toward reasoning tasks such as counting, while VLMs remain superior at object recognition—highlighting how alignment with language fundamentally reshapes multimodal capabilities.

CountBench, VG Attribution) grouped into 18 distinct capability categories, including relation understanding, object recognition, spatial understanding, robustness to corruption, and scene recognition. Results reveal MLLMs underperform their own vision backbone on perception tasks such as object recognition. On average, MLLMs trail their backbones by 7–10% across perception-oriented categories, with drops as large as 14% in standard object recognition and related benchmarks. Simultaneously, MLLMs demonstrate advantages in reasoning tasks such as counting and relational understanding, where language grounding provides complementary strengths. On average, MLLMs outperform their backbones by 20–25% in reasoning-focused capabilities, with gains exceeding 40% in some settings (e.g., spatial understanding, corruption robustness). These findings indicate that the integration of LLMs systematically boosts reasoning while degrading perception, underscoring the trade-offs introduced by multimodal alignment.

## 2 Methods

The evaluation of model performance was conducted using a standardized, comprehensive framework to ensure a fair and rigorous comparison between MLLMs and contrastive vision-language models. The methodology, including the selection of models and the evaluation datasets, is described in detail below.

### 2.1 Models and Capabilities

We included a wide range of MLLMs for evaluation, encompassing models from several leading families. The models were evaluated across multiple variants to provide a thorough assessment. The model families included Llama 4 [Touvron et al., 2023], LLaVA [Liu et al., 2024a, 2023a,b], Paligemma [Beyer et al., 2024, Steiner et al., 2024], Gemma [Team et al., 2024], Chameleon [Team, 2025], and Aya Vision [Dash et al., 2025]. In our analysis, we leveraged these design choices to reveal how different backbones, scales, and training regimes shape downstream performance on UniBench. For example, models such as LLaVA leverage CLIP-style encoders, while Paligemma and Gemma rely on integration with newer backbones such as SigLIP, which utilizes SoViT-400m architecture. These architectural pairings highlight the spectrum of strategies for bridging visual and textual modalities. The models also vary significantly in scale, ranging from lightweight 3B parameter variants optimized for efficiency, to mid-sized 7B–13B models balancing capacity with cost, and up to frontier-scale 27B+ systems designed for state-of-the-art performance.

The evaluation was performed using the UniBench framework [Al-Tahan et al., 2024b], a robust benchmark designed to assess a model's visual reasoning capabilities across a variety of tasks and benchmarks. We used all UniBench benchmarks, including well-known benchmarks such as Imagenet, CIFAR100, CLEVR, CountBench, and SugarCrepe, among many others, which span 18 capabilities: relations, standard object recognition, counting, spatial understanding, geographic diversity, specifies classification, depth estimation, pose detection, texture detection, satellite, character recognition, imagenet, natural transformations, rendition, challenging imagenet, corruption, medical, and scene recognition. For each benchmark in UniBench, we evaluated 5,000 samples per benchmark, and used the same subset across models.

## 2.2 Evaluating Multimodal LLLMs

The evaluation of MLLMs presents unique challenges beyond standard classification tasks. Since the models generate natural language outputs, a simple match-based evaluation is often insufficient to capture the nuance of the response. To address these challenges and provide a comprehensive analysis, we established several evaluation setups for our experiments. For our primary classification task, we tested multi-choice prompting but we also evaluated using other text based evaluation matching Appendix A. The task's parameters included a number of classes (4) and a prompt formatted as follows: *What type of object is in this photo? Choose one of the following options: class names.*
**Example:** *What type of object is in this photo? Choose one of the following options: (A) Apple (B) Dog (C) Cat (D) Orange.*

Beyond classification, we also evaluated the MLLMs ability to understand and reason about relationships between objects in image(s). For these relational understanding tasks, we evaluated a model's ability to correctly associate a given image with the correct caption from a set of captions. A correct overall response requires the model to have correctly associated the set captions with their images.

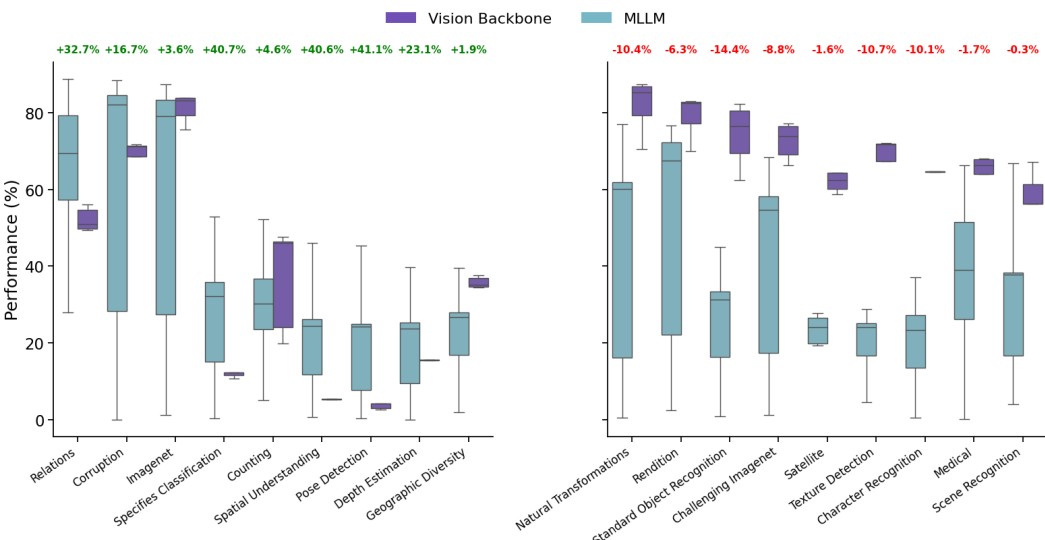

Figure 2: **Comparison of MLLMs and their vision backbones across vision-centric capabilities.** The x-axis denotes capability categories, with performance shown as box plots over multiple benchmarks. Numbers above each category indicate the relative difference in accuracy (%) between the best-performing MLLM and the best-performing vision backbone. The left panel highlights capabilities where MLLMs outperform vision backbones—primarily reasoning-oriented tasks such as relations, counting, and spatial understanding. The right panel shows capabilities where vision backbones outperform MLLMs, typically perception-heavy tasks such as object recognition, character recognition, and medical imaging.

# 3 Where do Multimodal LLMs excel or falter over VLMs

**MLLMs boost visual reasoning capabilities.** Figure 2 (left panel) shows that the average MLLM models demonstrated superior performance on a distinct set of capabilities, including Corruption, Counting, and various visual understanding tasks. These tasks require the model to go beyond simple feature extraction and engage in more complex processes like spatial understanding, robust perception under degradation, and the synthesis of abstract visual concepts.

The distribution of results across families highlights that this trend is not confined to a single architecture: models span includes Llava, PaliGemma, Gemma, and Llama Vision families on average shows gains when paired with an LLM. Importantly, the distributions reveal that the performance improvements of MLLMs are consistent over their vision-only backbones such as SigLIP, EVA, and CLIP.

**MLLMs offer worse perception than their own vision-backbones.** In contrast to their gains on reasoning tasks, MLLMs underperform their vision-backbones on perception capabilities, shown in Figure 2 (right panel). Tasks such as Natural Transformations, Rendition, Standard Object Recognition, Character Recognition, Medical, and Scene Recognition rely on fine-grained visual cues, low-level texture fidelity, and precise classification—all areas where vision encoders like SigLIP, EVA, and CLIP maintain a clear advantage.

**Instruction following offers a partial explanation of MLLMs perception weakness** In Figure 3, we evaluated instruction-following fidelity by measuring whether each model's outputs adhered to the expected multiple-choice response format. Compliance was scored using a rule-based validator that awarded full credit (score of 1) when a model produced exactly one valid option (e.g., A, (B), or **C**), partial credit (score of 0.5) when a valid choice was embedded in additional text, and no credit otherwise (score of 0).

Overall, performance varied widely across model families and sizes. The strongest models in this evaluation were Gemma 3 (27B) and Gemma 3 (4B), which achieved validity rates above 45%. In contrast, many smaller models and some instruction-tuned variants frequently deviated from the expected format, with validity rates dropping below 15%. Interestingly, while larger models generally showed better adherence, this was not uniform: several medium-scale models (e.g., Paligemma mixes and LLaVA variants) clustered around 35–40%, demonstrating only moderate reliability.

# 4 Practical Recommendations

To identify efficient proxies for full-scale evaluation, we analyzed benchmark correlations across both vision-language models (VLMs) and multimodal large language models (MLLMs). Tables 1 and 2 summarize the strongest correlations for each benchmark type and highlight curated subsets that can serve as representative evaluations.

In Table 1, object recognition (ImageNet-1k, 0.82) and Robustness (ImageNet-v2, 0.81) are highly correlated. These results suggest that traditional vision-oriented evaluations can be effectively approximated with smaller curated datasets. However, reasoning-oriented tasks are less predictable: Counting correlates moderately with CountBench (0.76), while Spatial reasoning is only weakly captured by DSPR Position (0.29).

In contrast, MLLMs display stronger correlations for multimodal and relational tasks, reflecting their broader training objectives. Object recognition is better captured by Cifar-100 (0.92) than by ImageNet-1k, and Spatial reasoning shows a much stronger correlation with DSPR Position (0.84) compared to VLMs. Relational reasoning also benefits from benchmarks like Flickr30k (0.74). Robustness remains well represented (ImageNet-v2, 0.98), but Counting again stands out as difficult to approximate, with a very low correlation to Clevr Count (0.16).

**Research opportunity for better multimodal alignment methods** MMLMs relatively worse performance on perception suggests room for better alignment methods to fully take advantage of the existing capabilities in the model's vision-backbone.

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

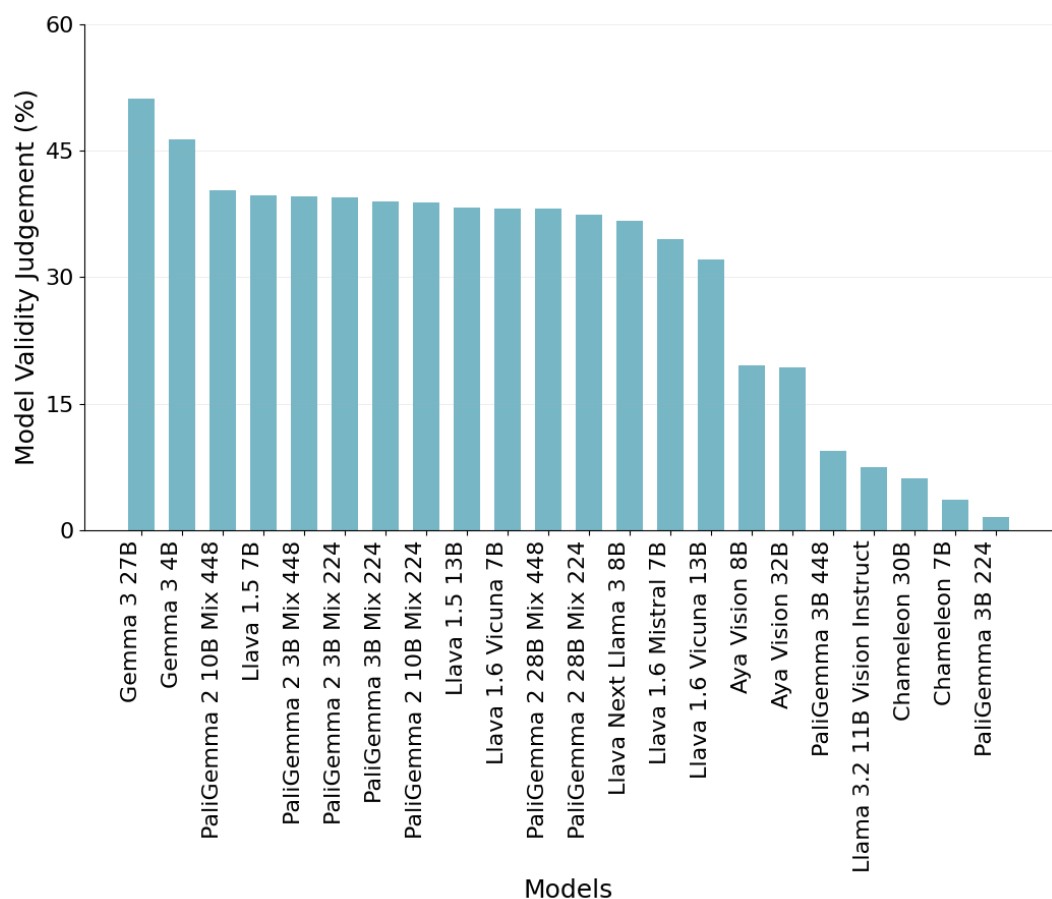

Figure 3: **Model validity judgement rates across evaluated models.** The percentage reflects how often each model produced outputs in the correct multiple-choice format (e.g., A, (B), **C**). Scores were computed by applying a rule-based validator that assigns correct for strictly valid outputs, 0.5 when only one valid answer is present but with extraneous text, and 0.0 otherwise. Models such as Gemma 3 (27B) and Gemma 3 (4B) showed the highest compliance, while smaller or instruction-tuned variants tended to generate less consistent formatting.

## A    Text Classification

For this task, the model was instructed to output the target class directly. The task's parameters included a number of classes (4) and a prompt formatted as follows: *What type of object is in this photo? Choose one from class names*. The evaluation for this task, which we term Text Classification, was a simple string-based check to determine if the target class name appeared anywhere in the MLLM's output text. While straightforward, this method has several drawbacks. For instance, the model's output might contain extraneous information, as an example, prompting Aya Vision 32B yeilds "The object in the photo is a number, specifically the numeral 7. Therefore, the correct answer is 4, as it represents a number." In other cases, the model may produce a valid but incorrect answer that is not one of the provided options, such as "Circle" or "Person" when the options were only MNIST classes. Another drawback is when the model might respond with the correct class but in a different format, such as "The object in the photo is a number seven," which would not be recognized by a strict string-matching algorithm.

## B    Additional results

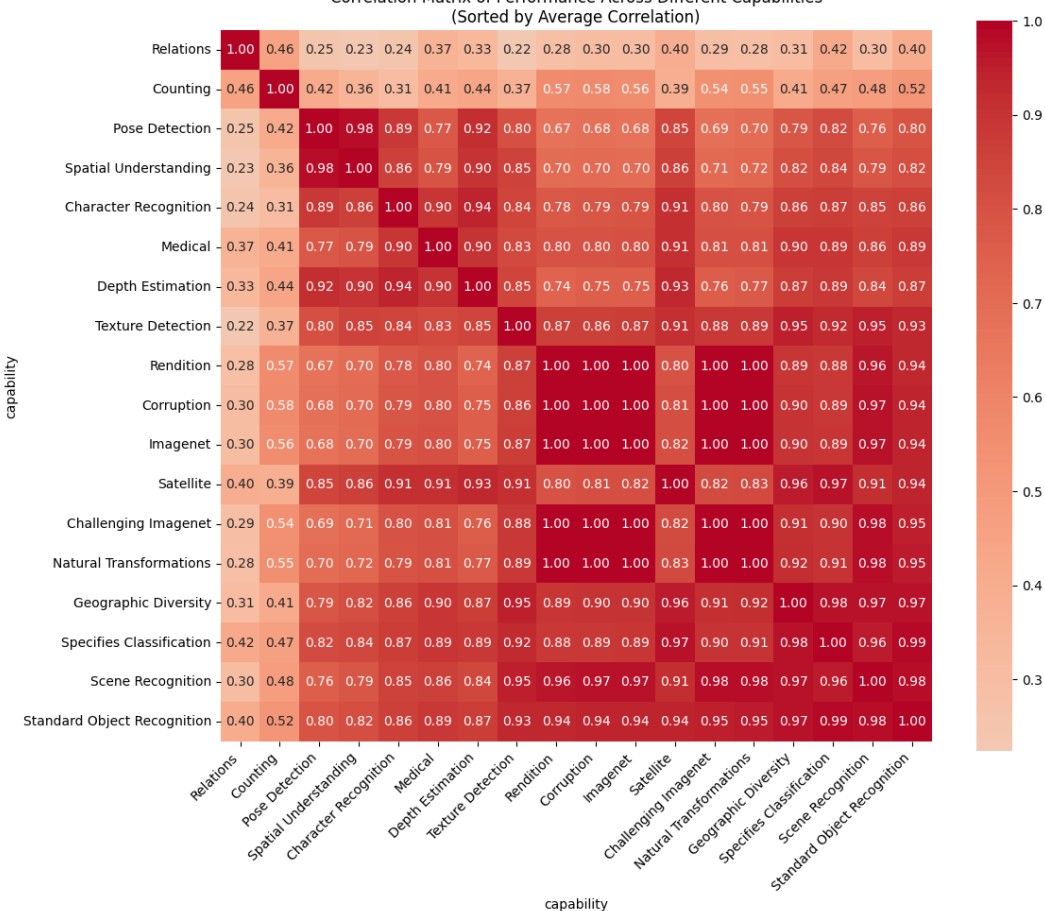

Figure 4: **Correlation matrix of model performance across different capabilities.** Warmer colors indicate stronger positive correlations, while cooler colors reflect weaker relationships. The results reveal that many visual recognition tasks—such as standard object recognition, scene recognition, and geographic diversity—are highly correlated, whereas tasks like relations and counting show weaker alignment. Overall, models share strengths across perceptual recognition tasks but diverge more on reasoning-oriented capabilities.

| Benchmark Type | Most Correlated Benchmark | Correlation Value |
|---|---|---|
| Object recognition | ImageNet-1k | 0.82 |
| Reasoning (Counting) | CountBench | 0.76 |
| Reasoning (Spatial) | DSPR Position | 0.29 |
| Relation | VG Attribution | 0.57 |
| Texture | DTD | 1 |
| Non-Natural Images | Resisc45 | 0.72 |
| Robustness | ImageNet-v2 | 0.81 |
| Corruption | ImageNet-c | 1 |

Table 1: **Evaluate on a curated list of benchmark types, to save time for VLMs.**

| Benchmark Type | Most Correlated Benchmark | Correlation Value |
|---|---|---|
| Object recognition | Cifar 100 | 0.92 |
| Reasoning (Counting) | Clevr Count | 0.16 |
| Reasoning (Spatial) | DSPR Position | 0.84 |
| Relation | Flickr30k | 0.74 |
| Texture | DTD | 1 |
| Non-Natural Images | Resisc45 | 0.84 |
| Robustness | ImageNet-v2 | 0.98 |
| Corruption | ImageNet-c | 1 |

Table 2: **Evaluate on a curated list of benchmark types, to save time for MLLMs.**

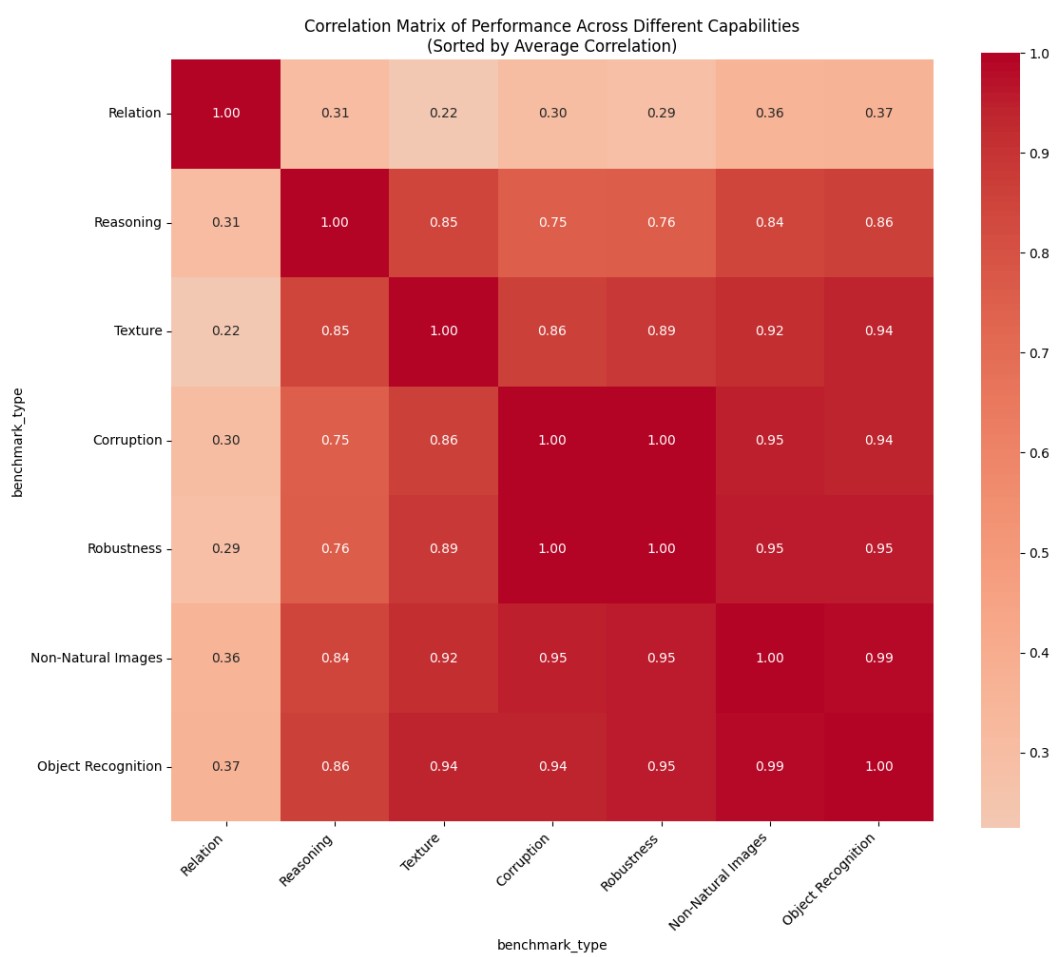

Figure 5:

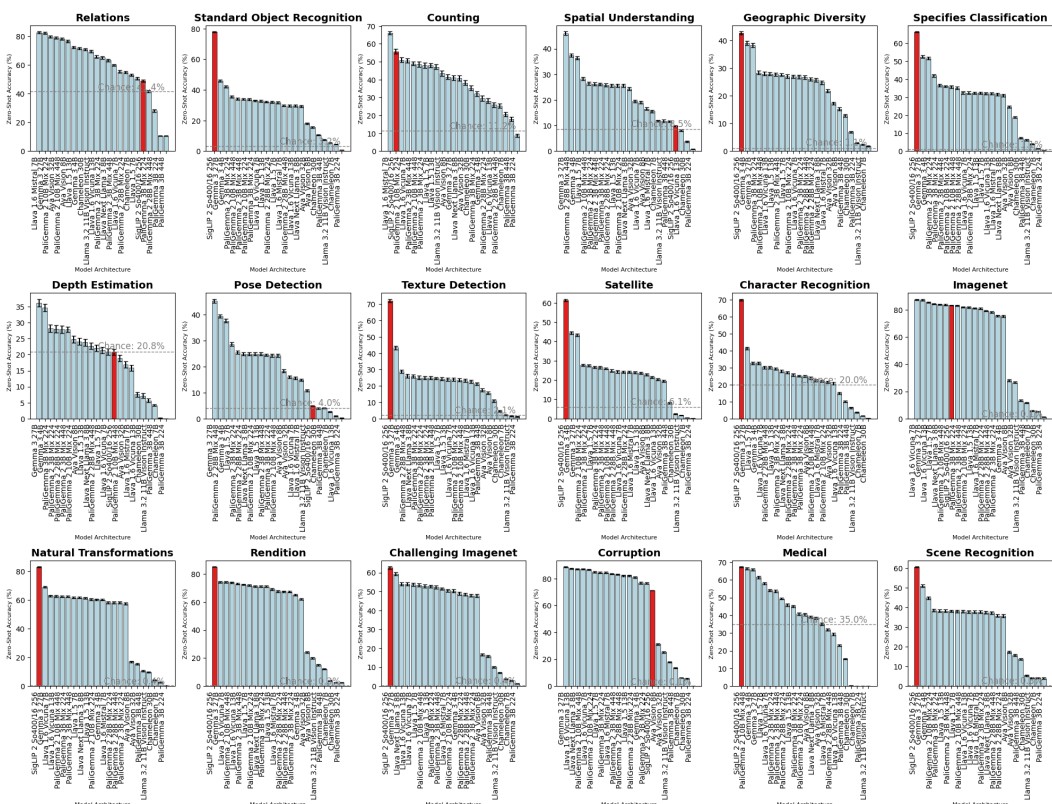

Figure 6: **Zero-shot evaluation of a wide range of model architectures across 18 visual understanding tasks.** While conventional vision encoders (e.g., SigLIP) achieve strong performance on recognition-heavy benchmarks (e.g., ImageNet, Object Recognition), VLMs consistently outperform them on higher-level reasoning tasks such as Relations, Counting, Spatial Understanding, and Geographic Diversity, highlighting their broader capability beyond standard recognition. Red bars highlight the performance of a reference model (SigLIP 2 So400m/14).

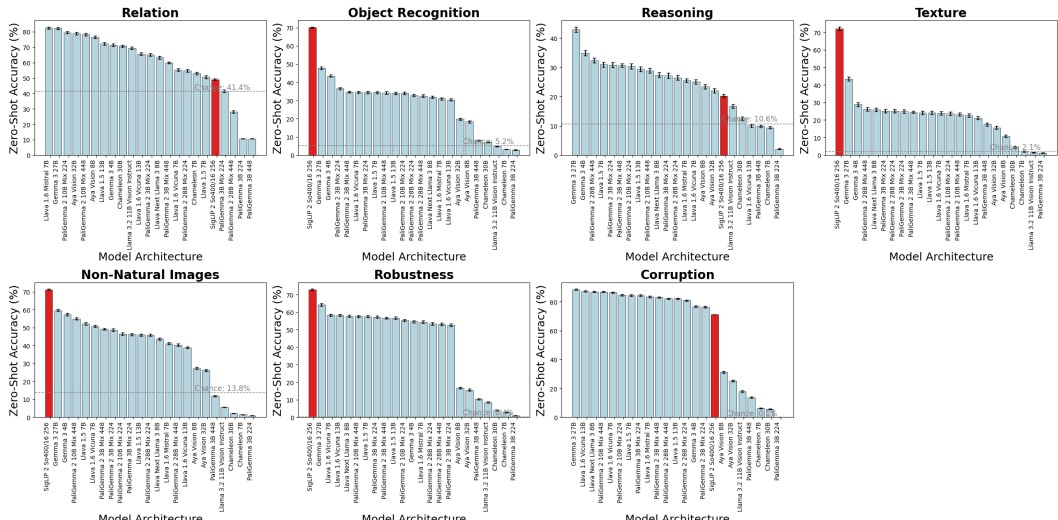

Figure 7: **Zero-shot performance of diverse model architectures across seven benchmark categories: Relation, Object Recognition, Reasoning, Texture, Non-Natural Images, Robustness, and Corruption.** VLMs consistently outperform unimodal vision encoders on reasoning-heavy tasks (e.g., Relation, Reasoning), while vision encoders such as SigLIP dominate low-level perception tasks (e.g., Texture, Object Recognition). These results highlight complementary strengths: VLMs excel in semantic and abstract understanding, whereas vision-only models remain strong in fine-grained recognition and robustness benchmarks. Red bars highlight the performance of a reference model (SigLIP 2 So400m/14).

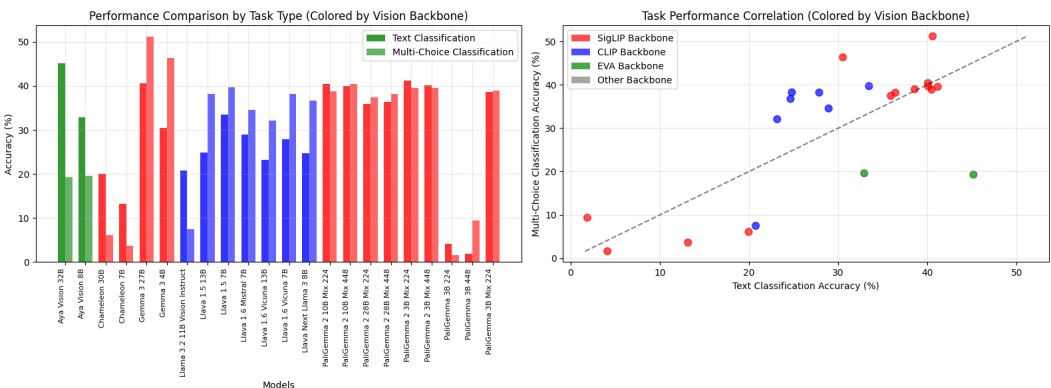

Figure 8: **(Left)** Performance comparison of different models on text classification (dark color) and multi-choice classification (light color) tasks. Models are grouped and colored by their vision backbone (see legend). **(Right)** Correlation between text and multi-choice classification task performance. Each point represents a model, and its color indicates the vision backbone used. The dashed gray line indicates the ideal scenario where performance on both tasks is equal. The figure suggests a positive correlation between performance on the two tasks, though with significant variability.

