# OpenReview forum: "Blinded by Language: Multimodal LLMs Underuse Their Vision Backbone"
_NeurIPS.cc/2025/Workshop/UniReps — UniReps2025_

### Official Review · Reviewer_Vcr9 · 2025-09-09
**Review of Blinded by Language: Multimodal LLMs Underuse Their Vision Backbone**

**Confidence:** 4

**Review:**

### Summary
This paper evaluates Multimodal Large Language Models (MLLMs) and their corresponding vision backbones (VLMs) on Unibench, a collection of 53 vision-language benchmarks, The authors demonstrate that MLLMs lag behind their own vision-backbones on core vision perception tasks. On the other hand, MLLMs perform significantly better than their VLM counterparts on reasoning tasks such as counting and spatial understanding. This demonstrates the benefit of the language capabilities of MLLMs in improving on such reasoning tasks. Finally, the authors provide a set of representative benchmarks for evaluating MLLMs.

### Strengths:
* The authors have performed an extensive evaluation, including multiple models across various scales.
* The main finding of the paper has been explained and backed with multiple results.

### Weaknesses:
* My major concern is that Table 1 is exactly the same as the Table 2 in the original Unibench paper, including all correlation values. How is this possible, despite both papers having a different set of models used for evaluation?
* In Section 2.1, the authors claim their analysis reveals how different backbones, scales and training regimes affect downstream performance on Unibench. The paper did not include any discussion regarding this, only a small section on the ability of these models in adhering to the output format. Nothing is mentioned about their actual performance on these benchmarks.
* The idea of having different representative benchmarks for MLLMs vs VLMs seems counterintuitive and biased. Ideally, all models should be evaluated on the same benchmark which is the one with the highest correlation. For example, when evaluating Object Recognition, there can only be one benchmark that captures this capability best and it should be independent of the model type.
* Presentation issues:
  * VLMs and vision-backbones are used interchangeably for no reason; in some sections the heading says VLMs while the text says vision-backbone.
  * The plots in Appendix are unreadable.

Overall, the paper lacks deeper analysis, and the findings remain descriptive rather than offering any in-depth insights.

**Score:**

2

**Topic Fit:**

2

---

### Official Review · Reviewer_ZjsP · 2025-09-12
**Interesting study; need to clarify perception vs reasoning distinction**

**Confidence:** 4

**Review:**

In this paper, the authors evaluate how MLLMs compare to their vision backbones on perception versus reasoning-oriented benchmarks. Using the UniBench framework spanning 18 visual capabilities, they find that MLLMs consistently underperform their vision backbones on perceptual tasks but achieve notable gains on reasoning tasks. This suggests that incorporating language may enhance reasoning at the expense of perceptual fidelity. The central finding, a potential trade-off between perception and reasoning, raises an important question for the UniReps community: is this trade-off grounded in modality (language vs. vision) or in some more specific computational trade-off?

Notes for clarification

- Perception vs reasoning split: the perception vs. reasoning split feels underspecified and, in some cases, mismatched with established cognitive science. For example, tasks such as counting and depth estimation are typically considered perceptual processes in humans. The authors should provide clearer definitions for how they classify tasks and justify why certain benchmarks (e.g., counting, depth) fall into reasoning rather than perception.

- Evaluation of backbones: the paper details how the MLLMs are evaluated on the Unibench subset but does not clearly explain how vision backbones were evaluated, particularly on reasoning-oriented benchmarks (e.g., CountBench, relational understanding). Since these models are not designed to produce natural language outputs, it would be helpful if the authors clarified whether they used probes, similarity scoring, or some other adaptation. Without this detail, it is difficult to assess the fairness of the backbone vs. MLLM comparisons.

- Typos: “MLLMs” in the title of Section 2.2, and in the first word of the “Research opportunity …” section under Practical Recommendations.

This is a well-motivated and timely paper investigating modality-driven trade-offs in multimodal models. It builds on and complements recent work (e.g., Fu, Bonnen, Guillory, & Darrell 2025) comparing vision language models to their vision encoders. With improvements in clarity on the two points above, the paper would make a strong contribution to the UniReps extended abstract track. Its discussion of perception vs. reasoning trade-offs in MLLMs is particularly well-suited to spark discussion at the workshop.

**Score:**

3

**Topic Fit:**

3

---

### Official Review · Reviewer_vnWp · 2025-09-15
**Review of Submission107**

**Confidence:** 4

**Review:**

**Summary:**

This paper systematically compares multimodal LLMs (MLLMs) with their underlying vision backbones across multiple benchmarks. It finds that while MLLMs gain clear advantages on reasoning-heavy tasks like counting and relational understanding, they consistently underperform their vision backbones on pure perception tasks such as object recognition. The authors attribute this gap partly to evaluation practices and propose a reduced benchmark suite to better assess the vision capabilities of MLLMs.

**Strengths:**

- The authors extensively benchmark MLLMs against their vision backbones across 48 datasets and 18 capability categories, providing a very comprehensive comparison;

- The paper is well-structured and clearly written, making the technical results easy to follow;

- It draws attention to the relatively underexplored but relevant problem of MLLMs underutilizing their vision backbone, a subtle limitation often ignored in the rush toward multimodal integration.

**Weaknesses:**

- The paper does not introduce any novel methodological contribution, as it is solely focused on evaluating the vision capabilities of MLLMs;

- The analysis of why MLLMs underperform on perception remains fairly shallow. The paper points to evaluation practices and issues related to instruction following but does not dig into deeper causes for this phenomenon.

**Score:**

3

**Topic Fit:**

2